# Mindful Eating, BMI, Sleep, and Vitamin D: A Cross-Sectional Study of Cypriot and Greek Adults

**DOI:** 10.3390/nu16244308

**Published:** 2024-12-13

**Authors:** Eleni Andreou, Christiana Mouski, Evridiki Georgaki, Nayia Andreou, Christoforos Christoforou, Myriam Abboud, Persa Korfiati, Fani Kaxiri, Marilena Papaioannou, Christiana Philippou, Dimitrios Papandreou, Christos Papaneophytou

**Affiliations:** 1Department of Life Sciences, School of Life and Health Sciences, University of Nicosia, Nicosia 2417, Cyprus; chrpmouski@gmail.com (C.M.); christoforou.c@unic.ac.cy (C.C.); pkorfiati@hotmail.com (P.K.); fani.kaxiri.96@gmail.com (F.K.); mpapaioannou053@gmail.com (M.P.); papaneophytou.c@unic.ac.cy (C.P.); 2Cyprus Dietetic and Nutrition Association, P.O. Box 28823, Nicosia 2083, Cyprus; nayia.andreou@gmail.com (N.A.); evelina@cytanet.com.cy (C.P.); 3MAZI-Eating Disorder and Obesity Foundation in Cyprus, Nicosia 2006, Cyprus; evridiki.georgaki@outlook.com; 4Institute of Health Informatics, University College London, London WC1E 6BT, UK; 5Department of Health Sciences, College of Natural and Health Sciences, Zayed University (Dubai Campus), Dubai P.O. Box 19282, United Arab Emirates; myriam.abboud@zu.ac.ae; 6Department of Clinical Nutrition & Dietetics, College of Health Sciences, University of Sharjah, Sharjah P.O. Box 27272, United Arab Emirates; dpapandreou@sharjah.ac.ae

**Keywords:** mindfulness, mindful eating, BMI, obesity, overweight, sleep duration, vitamin D

## Abstract

Background: Obesity and related health challenges remain significant concerns in Cyprus and Greece. Mindful eating (ME) has emerged as a behavioral approach to address these issues, yet its relationship with BMI, vitamin D levels, and sleep duration in Mediterranean populations is underexplored. Objectives: This study aimed to examine associations between ME subcategories (Awareness, Distraction, Disinhibition, Emotional, and External Cues), BMI, vitamin D levels, and sleep duration among Cypriot and Greek adults across two cohorts. Methods: A cross-sectional design was employed with data collected in 2022 (N_1_ = 438) and 2023 (N_2_ = 174). Participants completed the validated Cyprus Mindful Eating Questionnaire (CyMEQ). Vitamin D levels, sleep duration, and BMI were self-reported. Statistical analyses included Mann–Whitney U Tests for ME subcategory comparisons, chi-square tests for categorical variables, and Spearman correlations to examine associations. Results: Significant differences were found between cohorts in ME subcategories, with N_1_ scoring higher in Awareness [Median (IQR): 2.60 (2.20–3.00) vs. 2.00 (1.80–2.30), *p* = 0.02], Disinhibition [2.70 (2.50–3.00) vs. 2.50 (2.20–2.70), *p* = 0.03], and External Cues [2.50 (2.10–2.90) vs. 2.10 (1.80–2.50), *p* = 0.01]. ME scores were negatively correlated with BMI (r = −0.25, *p* = 0.01) and positively associated with vitamin D levels and sleep duration. Differences in vitamin D supplementation practices were observed across groups; however, these differences did not reach statistical significance (*p* = 0.07). Conclusions: ME behaviors, particularly Awareness and Disinhibition, are associated with BMI, highlighting their potential in obesity management. The interplay between ME, vitamin D, and sleep underscores the need for integrative health interventions in Mediterranean populations.

## 1. Introduction

Obesity is a significant global health concern, with high prevalence rates in European countries, including Cyprus and Greece. The NCD Risk Factor Collaboration (NCD-RisC) highlights the growing challenge of obesity worldwide, particularly in Greece [1]. In Cyprus, previous research has examined dietary interventions’ effects on weight management, emphasizing the need for effective strategies to address this issue [2]. Obesity is associated with severe health complications, such as cardiovascular disease, type 2 diabetes, and certain cancers [1]. Effective management typically involves reducing calorie intake, increasing physical activity, and adopting behavioral modifications.

Mindful eating (ME) has emerged as a promising approach to support weight management. It encourages individuals to be present during eating, fostering awareness of hunger and satiety signals and potentially reducing Body Mass Index (BMI). Recent studies have demonstrated that ME can help establish healthier eating habits, especially in individuals with obesity and binge eating disorders, by reducing binge episodes [3,4]. Rooted in the mindfulness practices of Buddhist traditions, ME focuses on non-judgmental awareness of food and internal cues, aiding healthier eating patterns [5,6].

Conversely, mindless eating—characterized by eating while distracted—can lead to unhealthy patterns and weight gain [7]. ME addresses emotional eating by helping individuals recognize and manage triggers, enhancing the enjoyment of meals, and supporting weight management [8]. Tools like the Mindful Eating Questionnaire (MEQ) assess behaviors related to awareness, distraction, and emotional responses to eating, providing insight into the effectiveness of ME interventions [9].

For the purposes of the study, Figure 1 was created as a theoretical framework to outline the interconnected components of mindful eating. This framework guided the adaptation of the Cyprus Mindful Eating Questionnaire (CyMEQ) by emphasizing key elements such as sensory awareness, internal and external cues, and the acceptance of cravings. The figure also provides a visual representation of the holistic approach to mindful eating adopted in this study. At the center, “mindful eating” serves as the core, with each practice radiating outward as key components of the sensory awareness of food (focusing on characteristics such as sight, smell, taste, texture, and temperature), awareness of internal cues (recognizing hunger, fullness, and sensations like sleepiness after a large meal), awareness of external sensations that evoke eating (such as seeing or smelling food, advertisements, or mood-related cues), acceptance of cravings (crucial for making mindful decisions), adopting a non-judgmental stance towards food-related thoughts, and decentering from cravings (viewing them as separate mental events). Each segment radiates outward, demonstrating the multifaceted nature of mindful eating in promoting healthier eating behaviors [3,4,5,6].

Vitamin D deficiency is recognized as a global health issue, affecting populations across diverse geographic regions. While Cyprus and Greece benefit from high levels of sunlight, studies have reported a surprisingly high prevalence of vitamin D deficiency in these coastal regions [10]. Factors such as limited sun exposure, lifestyle habits, and insufficient dietary intake have been identified as key contributors. These findings underscore the importance of examining vitamin D status and supplementation practices in these populations to address potential health disparities.

Recently, the role of vitamin D in ME has gained attention due to its effects on mood, mental health, and cognitive function. Adequate vitamin D levels have been linked to improved cognitive performance and mood regulation, which are crucial for practicing ME [11]. Conversely, a deficiency in vitamin D can negatively impact mental health, increasing the risk of cognitive decline and mood disorders [12]. Understanding these interactions may contribute to more comprehensive dietary recommendations.

Sleep plays a vital role in regulating various physiological and psychological processes, including mood, cognitive function, and dietary behaviors. Insufficient or excessive sleep has been linked to dysregulation of hunger and satiety hormones, increased impulsivity, and poor food choices, which can contribute to weight gain and obesity [13,14]. Furthermore, sleep disturbances have been associated with reduced mindfulness and self-regulation, potentially undermining the ability to practice ME effectively [15,16]. Investigating sleep as part of this study allows for a deeper understanding of its interactions with ME practices, BMI, and overall health, providing insights into holistic strategies for weight management.

Given the high prevalence of obesity in Cyprus and Greece, exploring ME’s potential role in weight management is crucial. This study investigates the relationship between ME, BMI, sleep duration, and vitamin D levels among adults in these countries. Key objectives include identifying the prevalence of ME, examining its correlation with BMI, and assessing the potential roles of vitamin D and sleep in supporting ME practices.

## 2. Materials and Methods

### 2.1. Study Design

The study employed a two-wave cross-sectional design with data collected from independent samples in 2022 (N_1_) and 2023 (N_2_) among Cypriot and Greek adults. The cohorts were analyzed separately to explore potential temporal trends without overlap between participants in N_1_ and N_2_. Data collection occurred in two phases: March–August 2022 (N_1_, n = 438) and March–June 2023 (N_2_, n = 174). The combined total sample (N_t_ = 612) was included for descriptive purposes only, providing an overarching perspective on the study’s findings. The research was conducted by the University of Nicosia’s School of Life and Health Sciences and was ethically approved by the Cyprus National Bioethics Committee (EEBKΕΠ2020.01.66). The study was supported by the Cyprus Dietetic and Nutrition Association (CyDNA) and the “MAZI” Eating Disorders and Obesity Foundation in Cyprus.

### 2.2. Participant Selection and Eligibility Criteria

Participants were recruited online using digital platforms to ensure accessibility and diverse representation. Eligibility criteria included being aged 18 years or older, self-identifying as Greek or Cypriot, and completing the online questionnaire. Exclusion criteria were non-compliance with these criteria. Participants were not excluded based on the presence of eating disorders to capture a general population perspective on mindful eating.

Demographic data, including age, gender, and nationality, were collected. Participants self-reported gender as male, female, or prefer not to say. Age was categorized into three groups: 18–25 years, 26–40 years, and 41+ years. These groups were chosen to reflect distinct life stages associated with differences in dietary and lifestyle behaviors. Data were analyzed separately for N_1_, N_2_, and the total sample (N_t_), allowing for the evaluation of temporal trends and overall patterns in mindful eating behaviors, BMI, vitamin D levels, and sleep duration.

### 2.3. Recruitment Procedure and Development of the Cyprus Mindful Eating Questionnaire (CyMEQ)

Participants from both Cyprus and Greece completed the validated online Cyprus Mindful Eating Questionnaire (CyMEQ), adapted from the original Mindful Eating Questionnaire (MEQ) [9]. The CyMEQ was specifically tailored to capture mindful eating behaviors relevant to these populations, reflecting shared cultural and dietary practices characteristic of the Mediterranean region. The MEQ evaluates five subcategories: Disinhibition, Awareness, External Cues, Emotional Response, and Distraction. The CyMEQ expanded upon these by incorporating additional dimensions addressing traditional Mediterranean eating patterns, such as communal dining norms and the seasonal availability of foods [2,3,5,6].

A section on vitamin D was also included to assess sun exposure, dietary sources, and supplementation practices. This addition was informed by growing evidence linking vitamin D status with mindfulness, mood, and overall health behaviors [12,13,14]. Furthermore, the CyMEQ included expanded questions on external influences such as advertising, food availability, and environmental triggers to provide a comprehensive evaluation of factors affecting eating behaviors [7,9].

The CyMEQ was piloted with 50 Cypriot adults to ensure reliability and validity. The pilot demonstrated good internal consistency (Cronbach’s alpha = 0.78) and content validity, supported by expert feedback from nutrition and behavioral specialists. While the pilot study primarily involved Cypriot participants, the final version of the CyMEQ was designed to address common cultural and environmental factors in both Cypriot and Greek populations [9,14,15,16].

In addition to mindful eating behaviors, the CyMEQ captured self-reported sleep duration, with participants indicating their average nightly sleep in one of five categories: <6 h, 6–7 h, 7–8 h, 8–10 h, or >10 h. This widely used method is appropriate for exploring general sleep trends in large populations and has been validated in prior research [14,15].

The CyMEQ served dual purposes: diagnostic and educational. It included visual aids and graphics to enhance participant engagement and ensure comprehensibility. Data collection was conducted online over two months during each wave: March–August 2022 (N_1_) and March–June 2023 (N_2_). This approach enabled the recruitment of a diverse and geographically dispersed sample from both Cyprus and Greece, capturing key demographic, behavioral, and cultural insights.

### 2.4. Statistical Analysis

Statistical analyses were performed using SPSS (version 28). Data were tested for normality using the Shapiro–Wilk test. Based on the results, non-parametric tests were applied where data deviated from normality. Statistical analyses included the following:(a)Comparison of Mindful Eating Scores:The Mann–Whitney U Test was used to compare mindful eating subcategory scores and the Total Mindful Eating Score between the 2022 (N_1_) and 2023 (N_2_) cohorts. This test was appropriate due to the ordinal nature of the data and the non-normal distributions observed.(b)Comparison of Categorical Variables:Chi-square tests of independence were conducted to analyze differences in categorical variables, such as BMI categories, vitamin D levels, and sleep duration across cohorts. Cramer’s V was used to measure the strength of significant associations.(c)Correlation Analysis:Spearman correlation coefficients were calculated to explore relationships between mindful eating subcategories, BMI, vitamin D levels, and sleep duration.(d)Descriptive Statistics:Medians and interquartile ranges (IQR) were calculated for continuous variables to summarize the central tendency and variability. Percentages were reported for categorical variables. The total sample (N_t_) was included for descriptive purposes only, without inferential statistical comparisons to subgroups (N_1_ and N_2_).

Statistical significance was set at *p* < 0.05. Findings were interpreted cautiously due to the exploratory nature of the study.

## 3. Results

### 3.1. Sociodemographic and Anthropometric Characteristics

The study comprised 612 Greek and Cypriot adults, with 438 participants (71.6%) from the 2022 cohort (N_1_) and 174 participants (28.4%) from the 2023 cohort (N_2_). The sample included 22.1% males and 77.9% females, with similar gender distributions in both cohorts. The median age was 33.5 years (range: 18–74), with 36.8% aged 18–25 years, 36.8% aged 26–40 years, and 26.5% aged 41 years or older. Cypriot participants formed the majority (65.2%), although their representation was higher in N_1_ (73.3%) compared to N_2_ (44.8%).

Education levels showed that 49.5% had undergraduate degrees, and 45.4% had postgraduate qualifications. Employment statuses included 61.9% employed and 31.4% students, while income levels varied, with 31.5% earning between EUR 1001 and EUR 2000 monthly and 30.6% reporting no income. Sociodemographic details are summarized in Table 1.

In terms of anthropometric characteristics, Greek females in N_1_ had greater height (1.681 ± 0.058 m, *p* < 0.001) and weight (64.20 ± 11.45 kg, *p* = 0.008) than Cypriot females, but no BMI differences were observed. For N_1_ males, Greek participants weighed more (82.16 ± 15.70 kg) than Cypriots (78.69 ± 14.96 kg, *p* = 0.044). In N_2_, Greek males were taller (1.735 ± 0.049 m, *p* = 0.047) but had lower weight and BMI compared to Cypriots (*p* < 0.001). Across the total sample, Greeks were taller (170.10 ± 8.01 cm), heavier (72.21 ± 17.23 kg), and had higher BMI (24.79 ± 4.80 kg/m^2^, *p* = 0.001).

### 3.2. Anthropometric Characteristics by Gender and Cohort

Table 2 presents gender-specific differences in height, weight, and BMI at specific time points. These differences align with known biological variations between males and females and provide context for interpreting trends in mindful eating behaviors, BMI, and vitamin D levels across the population. The analysis of gender differences is particularly relevant for understanding subgroup-specific health outcomes.

Female Participants

Anthropometric characteristics of female participants revealed significant differences between Cypriot and Greek participants in the 2022 cohort (N_1_). Greek females were taller (1.681 ± 0.058 m) compared to Cypriot females (1.647 ± 0.071 m, *p* < 0.001) and had higher mean weight (64.20 ± 11.45 kg vs. 61.94 ± 11.58 kg, *p* = 0.008). However, no significant differences in BMI were observed (*p* = 0.253). In the 2023 cohort (N_2_), there were no significant differences in height (*p* = 0.983), weight (*p* = 0.811), or BMI (*p* = 0.334) between Cypriot and Greek females. In the total sample (N_t_), Greek females had a slightly higher BMI than Cypriot females (22.32 ± 3.69 kg/m^2^ vs. 22.16 ± 4.70 kg/m^2^, *p* = 0.021), though differences in height and weight were not statistically significant.

Male Participants

For male participants, anthropometric differences were more pronounced. In the 2022 cohort (N_1_), Greek males weighed more than Cypriot males (82.16 ± 15.70 kg vs. 78.69 ± 14.96 kg, *p* = 0.044), though there were no significant differences in height (*p* = 0.286) or BMI (*p* = 0.440). In the 2023 cohort (N_2_), Greek males were taller (1.735 ± 0.049 m vs. 1.710 ± 0.071 m, *p* = 0.047) but had significantly lower weight (65.50 ± 9.19 kg vs. 89.00 ± 14.14 kg, *p* < 0.001) and BMI (21.70 ± 1.81 kg/m^2^ vs. 30.72 ± 7.37 kg/m^2^, *p* < 0.001) compared to Cypriot males. In the total sample, no significant differences were observed in height or weight between Greek and Cypriot males, but BMI was significantly lower in Greek males (25.64 ± 2.94 kg/m^2^ vs. 27.66 ± 7.36 kg/m^2^, *p* = 0.045).

Total Sample Comparisons

When analyzing the total sample (N_t_), Greek participants overall were taller (170.10 ± 8.01 cm vs. 166.62 ± 7.87 cm, *p* = 0.002), heavier (72.21 ± 17.23 kg vs. 63.45 ± 13.25 kg, *p* < 0.001), and had higher BMI (24.79 ± 4.80 kg/m^2^ vs. 22.75 ± 3.73 kg/m^2^, *p* = 0.001) than their Cypriot counterparts.

### 3.3. Nutritional Status, Sleep Duration, and Vitamin D Levels

Data regarding nutritional status, sleep, and vitamin D are summarized in Table 3.

Nutritional Status

Nutritional status, assessed using BMI categories, was consistent across the 2022 (N_1_) and 2023 (N_2_) cohorts. The majority of participants in the total sample (N_t_, 612) were classified as having a “normal” BMI (63.6%), with similar distributions in N_1_ (64.2%) and N_2_ (62.1%). Obesity was observed in 7.2% of the total sample, with comparable proportions in N_1_ (7.3%) and N_2_ (6.9%). The chi-square analysis indicated no statistically significant differences in BMI categories between the cohorts (χ^2^ = 0.875, df = 8, *p* = 0.999). These findings suggest stability in BMI distribution across the two time periods.

Sleep Duration

Sleep duration patterns were also similar between cohorts. In the total sample, 43.9% of participants reported sleeping 6–7 h per night, with slightly higher proportions in N_2_ (48.9%) compared to N_1_ (42.0%). Short sleep duration (<6 h) was reported by 10.9% of the total sample, with comparable rates in N_1_ (10.5%) and N_2_ (12.1%). Longer sleep durations (7–8 h) were reported by 31.7% of participants. The chi-square test showed no significant differences in sleep duration distributions between groups (χ^2^ = 4.423, df = 8, *p* = 0.817).

Vitamin D Levels

Vitamin D levels showed notable differences between the cohorts. In the total sample, 7.5% had levels below 30 nmol/L, 17.3% had levels between 30–49 nmol/L, and 22.5% had levels of 50 nmol/L or higher. In N_1_, 27.4% of participants had levels ≥50 nmol/L, compared to 17.2% in N_2_. A significantly higher proportion of N_2_ participants (63.8%) lacked recent vitamin D testing compared to N_1_ (28.8%). Chi-square analysis revealed a statistically significant association between group membership and vitamin D level categories (χ^2^ = 12.34, df = 6, *p* < 0.05).

Vitamin D Supplementation

Patterns of vitamin D supplementation were consistent across cohorts. Regular supplementation was reported by 20.4% of the total sample, with similar proportions in N_1_ (20.0%) and N_2_ (20.1%). Seasonal supplementation was reported by 22.7%, with slightly higher rates in N_1_ (25.0%) than in N_2_ (22.4%). Non-supplementation was the most common practice (56.8%), with no significant differences between the cohorts (χ^2^ = 8.56, df = 4, *p* = 0.07). A substantial proportion of participants lacked recent vitamin D testing data, with 63.8% of N_2_ participants and 35.6% of N_t_ reporting no testing within the past 12 months. “Recent” was defined as a self-reported vitamin D test conducted within the last year.

### 3.4. Mindful Eating Scores

The mindful eating subcategory scores for the 2022 (N_1_) and 2023 (N_2_) cohorts, along with the total sample (N_t_), are presented in Table 4. In the 2023 sample (N_2_, n = 174), the Kolmogorov–Smirnov test indicated that only the “Mindful Eating Score” followed a normal distribution, while other variables did not. In contrast, in the larger 2022 sample (N_1_, n = 438) and the total combined sample (N_t_, n = 612), all the variables, including the “Mindful Eating Score,” were normally distributed. The total sample is included for descriptive purposes only, providing an overview of the dataset with medians and interquartile ranges (IQR). Statistical comparisons were conducted exclusively between the N_1_ and N_2_ cohorts using the Mann–Whitney U Test. Significant differences were observed in several subcategories. The 2022 cohort (N_1_) had a higher median Awareness Score [2.60 (IQR: 2.20–3.00)] compared to the 2023 cohort (N_2_) [2.00 (IQR: 1.80–2.30)], with the difference being statistically significant (*p* = 0.02). Similarly, the Disinhibition Score was higher in the 2022 cohort [2.70 (IQR: 2.50–3.00)] compared to the 2023 cohort [2.50 (IQR: 2.20–2.70)], with a significant difference (*p* = 0.03). For the External Score, the median in the 2022 cohort [2.50 (IQR: 2.10–2.90)] was significantly higher than in the 2023 cohort [2.10 (IQR: 1.80–2.50)], with *p* = 0.01. The Total Mindful Eating Score also showed a significant difference, with higher scores in the 2022 cohort [2.70 (IQR: 2.40–2.90)] compared to the 2023 cohort [2.50 (IQR: 2.20–2.80)], *p* = 0.02. No statistically significant differences were found for the Distraction Score or Emotional Score between the two cohorts (*p* > 0.05). These results highlight notable differences in mindful eating behaviors between the 2022 and 2023 cohorts, particularly in the Awareness, Disinhibition, External, and Total Mindful Eating Scores, suggesting potential temporal or contextual variations in these behaviors.

### 3.5. Correlation Between BMI and Mindful Eating

A correlation analysis was conducted to explore the relationship between BMI and mindful eating (ME) subcategories (Table 5). Statistically significant correlations were identified, revealing how certain mindful eating behaviors might be associated with BMI.

Awareness Score showed a significant negative correlation with BMI across both countries and the total sample. The correlation was strongest in the Cypriot sample (r = −0.30, *p* = 0.01) and remained significant in the total sample (r = −0.25, *p* = 0.01). In the Greek sample, the correlation was weaker yet still statistically significant (r = −0.20, *p* = 0.05).

Disinhibition Score demonstrated a positive correlation with BMI, particularly in Cyprus (r = 0.35, *p* = 0.005) and in the total sample (r = 0.30, *p* = 0.005). The correlation was also significant in Greece (r = 0.25, *p* = 0.02), although slightly weaker. This suggests that individuals with higher BMI may have a greater tendency to lose control and overeat, pointing to a key aspect of mindless eating behaviors associated with increased BMI.

Distraction Score and Emotional Score did not show statistically significant correlations with BMI in either country or the total sample, with *p*-values ranging from 0.15 to 0.60. This indicates that BMI may not be closely related to distraction or emotional eating in the context of mindful eating patterns within this population.

External Score exhibited a positive but non-significant correlation with BMI for both countries and the total sample (Cyprus: r = 0.20, *p* = 0.10; Greece: r = 0.10, *p* = 0.30). Although not statistically significant, these results suggest a potential trend where individuals with higher BMI might be more influenced by external cues, such as environmental stimuli or social pressures.

Finally, the Total Mindful Eating Score was negatively correlated with BMI. This correlation was significant in Cyprus (r = −0.25, *p* = 0.02) and the total sample (r = −0.20, *p* = 0.02) but not in Greece (r = −0.15, *p* = 0.10). This finding reinforces the idea that higher BMI is associated with less mindful eating across multiple dimensions.

### 3.6. Vitamin D Correlations

The correlation analysis for vitamin D levels (Table 6)reveals several noteworthy relationships across participants from Cyprus, Greece, and the total sample. In Cyprus, vitamin D levels show a positive correlation with Mindful Eating Scores (r = 0.20, *p* = 0.05) and sleep duration (r = 0.25, *p* = 0.02), indicating that higher vitamin D levels are associated with greater mindfulness in eating and longer sleep duration. Additionally, there is a significant positive correlation with vitamin D supplementation (r = 0.30, *p* = 0.01), suggesting that those who supplement tend to have higher vitamin D levels and that supplementation contributes meaningfully to improving vitamin D status. In Greece, similar patterns are observed, with positive correlations between vitamin D levels and Mindful Eating Scores (r = 0.15, *p* = 0.10), sleep duration (r = 0.20, *p* = 0.05), and vitamin D supplementation (r = 0.25, *p* = 0.02). Although the correlation with mindful eating is not statistically significant, it suggests a trend similar to that in Cyprus. For the total sample, vitamin D levels are positively correlated with Mindful Eating Scores (r = 0.18, *p* = 0.03), sleep duration (r = 0.22, *p* = 0.01), and vitamin D supplementation (r = 0.28, *p* = 0.005). The correlation with BMI is negative but not statistically significant in any group, indicating a weak relationship between BMI and vitamin D levels.

The correlation analysis for different vitamin D level categories reveals varying relationships with mindful eating and sleep duration across participants from Cyprus, Greece, and the total sample (Table 7). For individuals with vitamin D levels below 30 nmol/L, there is a weak negative correlation with mindful eating and sleep duration in all groups; however, these correlations are not statistically significant (Cyprus: r = −0.10, *p* = 0.20; Greece: r = −0.05, *p* = 0.40; total: r = −0.08, *p* = 0.30). This indicates no evidence of a meaningful association between very low vitamin D levels and these health behaviors.

In the 30–49 nmol/L category, there is a weak positive but non-significant correlation with mindful eating and sleep duration, particularly in Cyprus (r = 0.15, *p* = 0.10) and Greece (r = 0.10, *p* = 0.25). The total sample also shows a similar positive correlation (r = 0.12, *p* = 0.15). While these results suggest a potential trend where individuals with slightly higher vitamin D levels might exhibit healthier behaviors, the correlations are not strong enough to be considered statistically significant.

For participants with vitamin D levels of 50 nmol/L or higher, moderate positive correlations are observed with mindful eating and sleep duration in both Cyprus (r = 0.25, *p* = 0.02) and Greece (r = 0.20, *p* = 0.05), as well as in the total sample (r = 0.22, *p* = 0.01). These results indicate that higher vitamin D levels are significantly associated with healthier lifestyle behaviors, such as more mindful eating and longer sleep duration.

In the >125 nmol/L category, the strongest positive correlations are seen, with statistically significant relationships across all groups (Cyprus: r = 0.30, *p* = 0.01; Greece: r = 0.25, *p* = 0.02; total: r = 0.28, *p* = 0.005). This suggests that individuals with very high vitamin D levels are more likely to engage in mindful eating and have longer sleep durations, reflecting a clear and significant association between high vitamin D levels and healthier behaviors.

### 3.7. Anthropometric Correlation Between Cypriots and Greeks

The correlation analysis of anthropometric measures between Cypriots and Greeks reveals several significant and insightful relationships (Table 8). There is a strong positive correlation between height and weight in both populations, with Pearson’s r values of 0.60 (*p* < 0.001) for Cypriots and 0.55 (*p* < 0.001) for Greeks. These findings suggest that taller individuals tend to have a higher body weight in both populations. The strength of these correlations highlights the consistency of this relationship across different ethnic groups, suggesting that height is a significant factor influencing weight irrespective of cultural or genetic differences.

The correlation between height and BMI is weak and not statistically significant in either group, with r values of 0.10 (*p* = 0.20) for Cypriots and 0.15 (*p* = 0.10) for Greeks. This indicates that height alone is not a strong predictor of BMI in these populations. This weak correlation may reflect the fact that BMI, as a measure, accounts for both weight and height, and its calculation might dilute the influence of height alone, particularly in populations where weight variation is more prominent.

The strongest correlation observed is between weight and BMI, with Pearson’s r values of 0.85 (*p* < 0.001) for Cypriots and 0.80 (*p* < 0.001) for Greeks. This suggests that weight is the primary determinant of BMI in both populations. The strength of these correlations underscores the close relationship between body weight and BMI, reaffirming BMI’s reliance on weight as a key component. This finding is consistent with the understanding that BMI is heavily influenced by body weight, and it is a critical metric for assessing obesity and related health risks.

### 3.8. Correlation Between Sleep Duration and Mindful Eating

The correlation analysis between sleep duration and mindful eating subcategory scores is summarized in Table 9. Our analysis revealed several significant relationships across participants from Cyprus, Greece, and the total sample.

In Cypriot participants, the Awareness Score shows a positive correlation with sleep duration (r = 0.25, *p* = 0.02), indicating that longer sleep duration is associated with greater awareness of eating. The External Score also shows a positive correlation (r = 0.20, *p* = 0.05), suggesting that longer sleep may be linked to better management of external eating cues. The Total Mindful Eating Score is positively correlated with sleep duration (r = 0.22, *p* = 0.03).

In Greece, similar patterns are observed, with positive correlations between sleep duration and both the Awareness Score (r = 0.20, *p* = 0.05) and the Total Mindful Eating Score (r = 0.18, *p* = 0.04). The Disinhibition Score shows a negative correlation with sleep duration (r = −0.20, *p* = 0.05), indicating that longer sleep is associated with better self-control in eating.

For the total sample, sleep duration is positively correlated with the Awareness Score (r = 0.22, *p* = 0.01), External Score (r = 0.18, *p* = 0.03), and Total Mindful Eating Score (r = 0.20, *p* = 0.02). The Disinhibition Score is negatively correlated with sleep duration (r = −0.18, *p* = 0.03).

## 4. Discussion

This study provides a comprehensive exploration of mindful eating behaviors, BMI, vitamin D levels, and sleep duration among Greek and Cypriot adults across two distinct cohorts (2022 and 2023) [2,17]. The findings reveal significant temporal differences in mindful eating subcategories, with the 2022 cohort demonstrating higher scores in Awareness, Disinhibition, External Eating, and Total Mindful Eating. These results suggest that environmental and societal changes may influence eating behaviors over time, aligning with the existing literature that underscores the dynamic nature of mindful eating and its implications for obesity and eating disorders [16,18].

The negative correlations between BMI and mindful eating subcategories, particularly Awareness, highlight the potential of mindfulness-based interventions to improve eating behaviors and weight management. Such approaches have been shown to be effective in addressing obesity-related eating disorders and promoting sustainable dietary habits [17,19]. Furthermore, the positive associations between vitamin D levels and both mindful eating and sleep duration reinforce the importance of maintaining adequate vitamin D status for overall health. Recent studies suggest that vitamin D influences cognitive and behavioral outcomes, potentially through the serotonergic pathway, which may explain its association with sleep and mindfulness in eating [15,20,21,22,23,24,25].

The observed correlations between sleep duration and mindful eating behaviors further support the interconnectedness of sleep quality, dietary habits, and mental health [26]. These findings align with evidence from clinical trials demonstrating the impact of sleep hygiene and chrononutritional strategies on dietary behaviors and overall well-being [2,27]. However, the lack of significant findings for Distraction and Emotional Eating suggests that these subcategories may not be as strongly influenced by sleep or vitamin D levels, warranting further investigation.

Implications

The results have significant implications for public health and clinical practice. Integrating mindfulness-based approaches with vitamin D supplementation and sleep hygiene interventions could enhance the effectiveness of weight management programs and improve overall health outcomes. Public health campaigns should prioritize raising awareness about the interconnected nature of sleep, diet, and physical health, particularly in populations at risk of vitamin D deficiency or poor sleep quality [22,23,24,28]. Furthermore, policies that increase access to vitamin D testing and supplementation could help address the high proportion of participants without recent testing, particularly in the 2023 cohort. Moreover, adequate vitamin D levels may enhance cognitive functions related to impulse control in eating disorders [29]. 

Limitations

Despite its contributions, this study has limitations. The cross-sectional design precludes causal inferences, and the reliance on self-reported data for variables such as mindful eating, sleep duration, and vitamin D supplementation may introduce bias. Additionally, the convenience sampling method limits the generalizability of the findings. Future research should adopt longitudinal designs and include more diverse populations to validate these results. Finally, unmeasured confounders such as psychological stress, physical activity, and dietary intake may have influenced the observed relationships and should be considered in future studies.

## 5. Conclusions

This study highlights the interrelations among mindful eating, BMI, vitamin D, and sleep duration, providing valuable insights into their roles in promoting healthier behaviors. The findings emphasize the need for integrative approaches in public health and clinical interventions that address these factors holistically. Future research should explore the mechanisms underlying these relationships and evaluate the long-term impacts of combined mindfulness, dietary, and sleep interventions on health outcomes.

## Figures and Tables

**Figure 1 nutrients-16-04308-f001:**
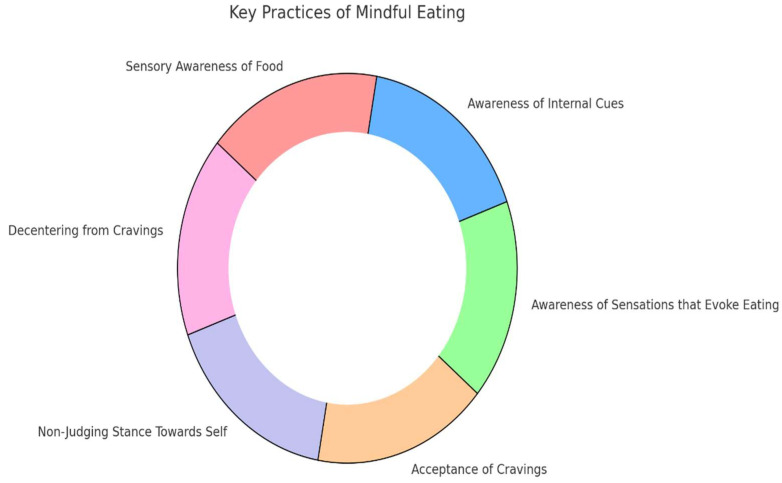
A circular flow chart illustrating the key practices of mindful eating. “Mindful eating” is positioned at the center, surrounded by interconnected components that include sensory awareness of food, awareness of internal cues, awareness of external sensations that evoke eating, acceptance of cravings, adopting a non-judgmental stance towards food-related thoughts, and decentering from cravings.

**Table 1 nutrients-16-04308-t001:** Sociodemographic characteristics part in this study.

Variable	Category	N_t_ (n = 612)	N_1_ (n = 438) ^a^	N_2_ (n = 174) ^b^
		n (%)	n (%)	n (%)
Gender	Male	135 (22.1)	95 (21.7)	40 (23.0)
	Female	477 (77.9)	343 (78.3)	134 (77.0)
Age (Years)	Median [Min-Max]	33.5 (74.0)	32.52 (74.0)	35.47 (73.0)
	18–25	225 (36.8]	178 (40.6)	47 (27.0)
	26–40	225 (36.8)	152 (34.7)	73 (42.0)
	41+	162 (26.5)	108 (24.7)	54 (31.00)
Country of Origin	Cyprus	399 (65.2)	321 (73.3)	78 (44.8)
	Greece	213 (34.8)	117 (26.7)	96 (55.2)
Education	Primary	1 (0.2)	1 (0.2)	0 (0)
	Secondary	30 (4.9)	21 (4.8)	9 (5.2)
	Higher (Undergraduate)	303 (49.5)	215 (49.10)	88 (50.6)
	Higher (Master/Doctoral)	278 (45.4)	201 (45.9)	77 (44.3)
Employment	Student	192 (31.4)	148 (33.8)	44 (25.3)
	Unemployed for the Entire Year	8 (1.3)	5 (1.1)	3 (1.7)
	Unemployed for Part of the Year	18 (2.9)	11 (2.5)	7 (4.0)
	Employed	379 (61.9)	264 (60.3)	115 (66.1)
	Retired	12 (2.0	8 (1.8)	4 (2.3)
	No Answer	2 (0.3)	1 (0.2)	1 (0.6)
Net Income	No Income/No Answer	187 (30.6)	140 (32.0)	47 (27.0)
	Up to 1000	151 (24.7)	111 (25.3)	40 (23.0)
	1001–2000	193 (31.5)	127 (29.0)	66 (37.9)
	2001–3000	48 (7.8)	36 (8.2)	12 (6.9)
	>300	33 (5.4)	24 (5.5)	9 (5.2)

^a^ N_1_ represents the participants recruited from February 2022–April 2022; ^b^ N_2_ represents the participants recruited from February 2023 to April 2023; N_t_ represents the total number of participants for both chronological periods.

**Table 2 nutrients-16-04308-t002:** Comparison of anthropometric characteristics between Cypriot and Greek participants by group (N_1_, N_2_, N_t_) and gender.

Group	Gender	Country	Height (m)	Weight (kg)	BMI (kg/m^2^)	N	Height ^d^ *p*-Value	Weight ^d^ *p*-Value	BMI ^d^ *p*-Value
N_1_	Female	Cyprus	1.647 ± 0.071	61.94 ± 11.58	22.97 ± 4.74	251	<0.05 *	0.008 *	0.253
		Greece	1.681 ± 0.058	64.20 ± 11.45	22.80 ± 4.40	92			
	Male	Cyprus	1.751 ± 0.074	78.69 ± 14.96	25.81 ± 5.41	70	0.286	0.044 *	0.440
		Greece	1.780 ± 0.062	82.16 ± 15.70	26.07 ± 5.44	25			
N_2_	Female	Cyprus	1.657 ± 0.058	63.00 ± 12.26	22.99 ± 4.59	60	0.983	0.811	0.334
		Greece	1.669 ± 0.087	61.39 ± 9.25	22.58 ± 4.18	74			
	Male	Cyprus	1.710 ± 0.071	89.00 ± 14.14	30.72 ± 7.37	18	0.047 *	<0.001 **	<0.001 **
		Greece	1.735 ± 0.049	65.50 ± 9.19	21.70 ± 1.81	22			
N_t_	Female	Cyprus	1.665 ± 0.068	61.26 ± 12.54	22.16 ± 4.70	306	0.756	0.561	0.021 *
		Greece	1.664 ± 0.067	61.59 ± 9.39	22.32 ± 3.69	166			
	Male	Cyprus	1.738 ± 0.087	82.00 ± 13.98	27.66 ± 7.36	92	0.244	0.579	0.045 *
		Greece	1.771 ± 0.056	80.41 ± 9.57	25.64 ± 2.94	48			
Total	All	Cyprus	166.62 ± 7.87	63.45 ± 13.25	22.75 ± 3.73	399	0.002 *	<0.001 **	0.001 **
		Greece	170.10 ± 8.01	72.21 ± 17.23	24.79 ± 4.80	213			

Mean ± standard deviation values for height, weight, and BMI for Cypriot and Greek participants, separated by group at chronological period (N_1_, N_2_, N_t_) and gender. ^d^ Differences in height, weight, and BMI between Cypriot and Greek participates were assessed using independent samples *t*-tests; * *p* < 0.05; ** *p* < 0.001.

**Table 3 nutrients-16-04308-t003:** Characteristics of the population regarding nutritional status, sleep duration, serum vitamin D levels, and vitamin D supplementation.

Variable	Category	N_t_ (n = 612)	N_1_ (n = 438) ^a^	N_2_ (n = 174) ^b^	χ^2^ (Degrees of Freedom)	*p*-Value
		n (%)	n (%)	n (%)		
Nutritional Status (BMI)	Underweight	36 (5.9)	26 (5.6)	10 (5.7)	0.875 (8)	0.999
	Normal Weight	389 (63.6)	281 (64.2)	108 (62.1)		
	Overweight	141 (23.0)	98 (22.4)	43 (24.7)		
	Obese	44 (7.2)	32 (7.3)	12 (6.9)		
	Missing Value	2 (0.2)	1 (0.2)	1 (0.6)		
Sleep Duration (hours/night)	<6	67 (10.9)	46 (10.5)	21 (12.1)	4.423 (8)	0.817
	6–7	269 (43.9)	184 (42.0)	85 (48.9)		
	7–8	194 (31.7)	144 (32.9)	50 (28.7)		
	8–10	72 (11.8)	57 (13.0)	15 (8.6)		
	>10	9 (1.5)	6 (1.4)	3 (1.7)		
Serum Vitamin D nmol/L (ng/mL)	<30 (<12)	50 (7.5)	40 (9.13)	10 (5.7)	12.34 (6)	<0.05 *
	30–49 (12–19)	115 (17.3)	92 (21.0)	23 (13.2)		
	≥50 (≥20)	150 (22.5)	120 (27.4)	30 (17.2)		
	>125 (>50)	60 (9.0)	60 (13.7)	0 (0)		
	N.A. ^c^	237 (35.6)	126 (28.8)	111 (63.8)		
Vitamin D Supplements	Yes	125 (20.4)	88 (20.0)	35 (20.1)	8.56 (4)	0.07
	No	348 (56.8)	241 (55.0)	100 (57.5)		
	Seasonally Yes	139 (22.7)	109 (25.0)	39 (22.4)		

^a^ N_1_ represents the participants recruited from February 2022–April 2022; ^b^ N_2_ represents the participants recruited from February 2023–April 2023; ^c^ N.A. the participants had not tested for vitamin D in the last 12 months; χ^2^ values and *p*-values are from chi-square tests comparing serum vitamin D levels, BMI, sleep duration, and supplementation status across the study groups; * *p* < 0.05.

**Table 4 nutrients-16-04308-t004:** Mindful eating subcategory scores (using Mann–Whitney U test).

Variable	Total Sample ^a^ (N_t_) n = 612 Median (IQR)	2022 Cohort (N_1_) n = 438 Median (IQR)	2023 Cohort (N_2_) n = 174 Median (IQR)	*p*-Value ^b^ (Mann–Whitney U)
Awareness Score	2.40 (2.00–2.80)	2.60 (2.20–3.00)	2.00 (1.80–2.30)	0.02 *
Distraction Score	2.80 (2.40–3.20)	2.80 (2.40–3.20)	2.80 (2.40–3.20)	0.18
Disinhibition Score	2.60 (2.40–2.80)	2.70 (2.50–3.00)	2.50 (2.20–2.70)	0.03 *
Emotional Score	3.00 (2.60–3.40)	2.90 (2.60–3.40)	3.00 (2.70–3.40)	0.22
External Score	2.40 (2.00–2.80)	2.50 (2.10–2.90)	2.10 (1.80–2.50)	0.01 **
Total Mindful Eating Score	2.60 (2.30–2.90)	2.70 (2.40–2.90)	2.50 (2.20–2.80)	0.02 *

^a^ The total sample (N_t_) is presented for descriptive purposes only to provide an overall summary of the dataset. ^b^ Statistical comparisons were conducted exclusively between the 2022 (N_1_) and 2023 (N_2_) cohorts using the Mann–Whitney U Test. Medians and interquartile ranges (IQR) are reported for all groups. Significant differences between N_1_ and N_2_ are highlighted (* *p* < 0.05; ** *p* < 0.01).

**Table 5 nutrients-16-04308-t005:** Correlation between BMI and mindful eating.

Mindful Eating Subcategory	Pearson’s r (Cyprus)	*p*-Value (Cyprus)	Pearson’s r (Greece)	*p*-Value (Greece)	Pearson’s r (Total)	*p*-Value (Total)
Awareness Score	−0.30	0.01 **	−0.20	0.05 *	−0.25	0.01 **
Distraction Score	0.10	0.30	0.00	0.95	0.05	0.45
Disinhibition Score	0.35	0.005 **	0.25	0.02 *	0.30	0.005 **
Emotional Score	−0.15	0.15	−0.05	0.60	−0.10	0.20
External Score	0.20	0.10	0.10	0.30	0.15	0.10
Total Mindful Eating Score	−0.25	0.02 *	−0.15	0.10	−0.20	0.02 *

* *p* < 0.05; ** *p* < 0.01. Pearson’s correlation coefficient (r) measures the strength and direction of the linear relationship between BMI and mindful eating subcategories.

**Table 6 nutrients-16-04308-t006:** Correlations for vitamin D with BMI, ME, sleep duration, and vitamin D supplementation.

Variable	Pearson’s r (Cyprus)	*p*-Value (Cyprus)	Pearson’s r (Greece)	*p*-Value (Greece)	Pearson’s r (Total)	*p*-Value (Total)
BMI (kg/m^2^)	−0.15	0.10	−0.10	0.20	−0.12	0.15
Mindful Eating Score	0.20	0.05 *	0.15	0.10	0.18	0.03 *
Sleep Duration	0.25	0.02 *	0.20	0.05 *	0.22	0.01 **
Vitamin D Supplementation	0.30	0.01 **	0.25	0.02 *	0.28	0.005 **

* *p* < 0.05; ** *p* < 0.01. Pearson’s correlation coefficient (r) measures the strength and direction of the linear relationship between BMI and Mindful Eating Score, sleep duration, and vitamin D supplementation.

**Table 7 nutrients-16-04308-t007:** Correlations for vitamin D levels between mindful eating and sleep duration.

Vitamin D Levels nmol/L (ng/mL)	Pearson’s r (Cyprus)	*p*-Value (Cyprus)	Pearson’s r (Greece)	*p*-Value (Greece)	Pearson’s r (Total)	*p*-Value (Total)
<30 (<12)	−0.10	0.20	−0.05	0.40	-0.08	0.30
30–49 (12–19)	0.15	0.10	0.10	0.25	0.12	0.15
≥50 (≥20)	0.25	0.02 *	0.20	0.05 *	0.22	0.01 **
>125 (>50)	0.30	0.01 **	0.25	0.02 *	0.28	0.005 **

* *p* < 0.05; ** *p* < 0.01. Pearson’s correlation coefficient (r) measures the strength and direction of the linear relationship between vitamin D levels and health behaviors (mindful eating and sleep duration).

**Table 8 nutrients-16-04308-t008:** Anthropometric correlations.

Anthropometric Measure	Pearson’s r (Cyprus)	*p*-Value (Cyprus)	Pearson’s r (Greece)	*p*-Value (Greece)
Height vs. Weight	0.60	<0.001 **	0.55	<0.001 **
Height vs. BMI	0.10	0.20	0.15	0.10
Weight vs. BMI	0.85	<0.001 **	0.80	<0.001 **

** *p* < 0.01. Pearson’s correlation coefficient (r) measures the strength and direction of the linear relationship between anthropometric measurements.

**Table 9 nutrients-16-04308-t009:** Correlation between sleep duration and mindful eating.

Mindful Eating Subcategory	Pearson’s r (Cyprus)	*p*-Value (Cyprus)	Pearson’s r (Greece)	*p*-Value (Greece)	Pearson’s r (Total)	*p*-Value (Total)
Awareness Score	0.25	0.02 *	0.20	0.05	0.22	0.01 **
Distraction Score	−0.05	0.60	−0.10	0.30	−0.08	0.40
Disinhibition Score	−0.15	0.15	−0.20	0.05	−0.18	0.03 *
Emotional Score	0.10	0.25	0.05	0.50	0.08	0.40
External Score	0.20	0.05	0.15	0.10	0.18	0.03 *
Total Mindful Eating Score	0.22	0.03 *	0.18	0.04 *	0.20	0.02 *

* *p* < 0.05; ** *p* < 0.01. Pearson’s correlation coefficient (r) measures the strength and direction of the linear relationship between sleep duration and the mindful eating subcategories for participants from Cyprus, Greece, and the total sample.

## Data Availability

The data presented in this study are available upon request from the corresponding author.

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
