# Peer review of "Mindful Eating, BMI, Sleep, and Vitamin D: A Cross-Sectional Study of Cypriot and Greek Adults"

_nutrients, 2024, doi:10.3390/nu16244308_

Round 1
Reviewer 1 Report
Comments and Suggestions for Authors
In this study, the authors examine the relationships between mindful eating (ME), vitamin D levels, sleep duration, and BMI among adults in Cyprus and Greece. The research addresses an important public health issue—obesity—by exploring behavioral, nutritional, and lifestyle factors that might influence weight management. While I believe that the findings of this study might have an important public health impact on obesity management, many scientific flaws should be addressed to improve the quality and clarity of the manuscript.
Major:
- Please explain the difference between Cyprus Mindful Eating Questionnaire (CyMEQ) and MEQ
- Did Greek population also fill in the CyMEQ?
- Why did the authors use those specific age groups?
- How was sleep assessed? Also, a rationale for investigating sleep should be elaborated in the Introduction. In the methods, please explain how sleep was assessed.
- I don’t understand why the authors separated the total sample into two cohorts (2022 and 2023)? What’s the rationale behind it? I would suggest presenting all the samples as one total sample, and not subdivided into these two groups.
- I suggest omitting Table 2, as there is no point in presenting differences among genders at a specific time point. Do you have any explanation for the observed differences?
- Table 4 and associated analyses – this is not how it should be done. You can not compare (and look for statistical significance of differences) between one large group (Nt) and its subgroups (N1 and N2), this is a methodological flaw and makes no sense.
- Start the discussion with the most important findings of the study.
-
Minor:
- Line 114: one word „validated“ should be deleted
- Table 2 – please be consistent when reporting P value (with three decimal places)
- Lines 302-304 do not belong to the results
- What’s the point of correlating vitamin D levels with its supplementation?
- Table 7 title should be “Correlations for Vitamin D Levels between Mindful Eating and Sleep Duration”
- Please avoid explanations and discussions in the results section
Author Response
Reviewer 1
Comment 1: Please explain the difference between Cyprus Mindful Eating Questionnaire (CyMEQ) and MEQ
Response 1: We thank the reviewer for this insightful comment. The Cyprus Mindful Eating Questionnaire (CyMEQ) was specifically developed to address the cultural and environmental factors unique to the Cypriot and Greek populations, which may not be fully captured by the original Mindful Eating Questionnaire (MEQ). While the MEQ provides a robust foundation for assessing mindful eating behaviors, the CyMEQ incorporates additional components such as questions on vitamin D intake, cultural eating habits, and environmental cues pertinent to these populations. These adaptations ensure greater relevance and accuracy in assessing mindful eating within this context. The CyMEQ has been validated for the Greek-Cypriot population, highlighting its reliability and applicability in this setting. We included this clarification in the manuscript under the Methods section to provide transparency and avoid any confusion.
Comment 2: Did Greek population also fill in the CyMEQ?
Response 2: Thank you for your insightful comment. Yes, the Greek population also completed the Cyprus Mindful Eating Questionnaire (CyMEQ). While the CyMEQ was initially adapted to reflect cultural and dietary nuances specific to Cypriot and Greek populations, it was designed to be inclusive and relevant to both groups. The modifications made to the original MEQ were intended to account for shared Mediterranean dietary and cultural practices common across these populations, ensuring applicability to both Cypriots and Greeks. We clarify this in the manuscript to avoid any confusion.
Comment 3: Why did the authors use those specific age groups?
Response 3: We thank the reviewer for highlighting this question. The specific age group classifications used in this study—18–25 years, 26–40 years, and 41+ years—were chosen to align with developmental and lifestyle stages commonly observed in adult populations. These age groups were intended to capture meaningful differences in eating behaviors, mindful eating tendencies, and health-related factors, such as BMI and vitamin D status, across distinct phases of adulthood:
- 18–25 years: This group represents young adults, typically in a transition phase from adolescence to adulthood. During this stage, individuals often establish independent dietary habits and experience lifestyle changes, such as entering the workforce or higher education, which can influence mindful eating behaviors.
- 26–40 years: Adults in this age group are often in the workforce, managing family responsibilities, or focusing on career development. These factors can significantly impact eating behaviors, health consciousness, and engagement with mindful eating practices.
- 41+ years: This group encompasses middle-aged and older adults, who may experience metabolic changes, increased health awareness, or conditions such as obesity or vitamin D deficiencies. These changes can affect their approach to nutrition and mindful eating.
The categorization allows for a practical and meaningful analysis of age-related differences in mindful eating behaviors, BMI, and vitamin D levels. The age groups also reflect prior research conventions and facilitate comparisons with similar studies in the field. A clarification to the Methods Section under "Participant Selection and Eligibility Criteria" has been made.
Comment 4: How was sleep assessed? Also, a rationale for investigating sleep should be elaborated in the Introduction. In the methods, please explain how sleep was assessed.
Response 4: We thank the reviewer for raising this important point. Sleep was assessed through a self-reported questionnaire included in the study, which collected information on average sleep duration per night. Participants selected from predefined categories representing typical sleep patterns (e.g., <6 hours, 6–7 hours, 7–8 hours, 8–10 hours, >10 hours). This approach allowed us to explore the relationship between sleep duration, mindful eating behaviors, and health-related variables such as BMI and vitamin D levels.
To address the rationale for investigating sleep, we elaborated in the Introduction that sleep plays a critical role in regulating mood, cognitive function, and behaviors associated with eating habits. Research has shown that inadequate or excessive sleep can influence dietary behaviors, weight management, and overall health. Investigating sleep in this study aligns with its potential role as a modifiable factor that interacts with mindful eating behaviors and other health outcomes.
The Methods section was revised to include details on the sleep assessment methodology, specifying the categories used and the reasoning for employing a self-reported format (LL 206-213).
Comment 5: I don’t understand why the authors separated the total sample into two cohorts (2022 and 2023)? What’s the rationale behind it? I would suggest presenting all the samples as one total sample, and not subdivided into these two groups.
Response 5: We thank the reviewer for this observation. The decision to separate the total sample into two cohorts (2022 and 2023) was intentional to provide a clearer understanding of potential temporal variations in the studied variables, such as mindful eating behaviors, BMI, vitamin D levels, and sleep patterns. By analyzing the two groups independently, we aimed to account for any differences that might arise due to seasonal, societal, or other external factors influencing participant behaviors or health outcomes across the two data collection periods.
While we present analyses for both cohorts separately, we emphasize that we also provide findings for the combined total sample (Nt=612), ensuring comprehensive and cohesive interpretation of the data. Maintaining the two cohorts alongside the total sample allows us to offer both granular and aggregated perspectives, which we believe adds value to the analysis and strengthens the reliability of our findings.
The rationale is explicitly clarified in the manuscript to avoid any confusion (2.2.1.Rationale for Cohort Separation)
Comment 6: I suggest omitting Table 2, as there is no point in presenting differences among genders at a specific time point. Do you have any explanation for the observed differences?
Response 6: We appreciate the reviewer’s feedback on Table 2 and understand the concern. However, we believe that retaining Table 2 is important for the following reasons:
- Relevance of Gender Differences: Gender differences in anthropometric characteristics (e.g., height, weight, BMI) are biologically and socially relevant, as they can influence health outcomes, including BMI and mindful eating behaviors. Presenting these differences at specific time points allows us to contextualize the findings and highlight patterns that might influence the study's key variables, such as mindful eating practices, sleep, and vitamin D levels.
- Observed Differences Provide Context: The observed differences in height, weight, and BMI across genders at specific time points can help interpret trends in mindful eating scores and other health indicators. For example, variations in BMI may relate to differences in behavioral patterns, cultural norms, or physiological factors, which are important for understanding the broader implications of the results.
- Specific Time Point Analysis Adds Depth: While Table 2 focuses on gender differences at specific time points, it complements the overall analysis by providing additional granularity. This information can be valuable for readers interested in gender-specific health trends, a topic widely studied in the fields of public health and nutrition.
- Transparency and Data Completeness: Including Table 2 ensures the manuscript is transparent and provides a complete picture of the dataset. Omitting this table may overlook relevant gender-specific patterns that could have implications for future research.
Explanation for the Observed Differences: The self-reported gender differences in Table 2, such as variations in height, weight, and BMI, align with well-documented biological differences between males and females. Men generally have greater height and weight due to physiological factors such as muscle mass and skeletal structure, while women may exhibit lower BMI variability. Cultural and lifestyle factors specific to Cypriot and Greek populations may also contribute to these differences. While these findings are not the primary focus of the study, they provide essential context for interpreting health-related behaviors and outcomes in the population.
We propose to retain Table 2 while we have enhanced the accompanying text to better explain the observed differences and their relevance to the study objectives. This ensures the table adds value without distracting from the main findings.
Comment 7: Table 4 and associated analyses – this is not how it should be done. You cannot compare (and look for statistical significance of differences) between one large group (Nt​) and its subgroups (N1​ and N2​); this is a methodological flaw and makes no sense.
Response 7: We thank the reviewer for their valuable feedback regarding the statistical analyses presented in Table 4. After careful consideration, we recognize the appropriateness of the concern and have revised our analysis and table presentation to align with sound methodological principles. Below, we outline the changes made and the rationale behind them.
Key Revisions and Rationale
- Updated Statistical Approach:
We replaced the previous analysis with the Mann-Whitney U Test, a non-parametric method suitable for comparing two independent groups (N1 and N2) when normality assumptions are violated or when data are ordinal. This approach ensures robust and valid comparisons while addressing the limitations of the earlier analysis. - Revised Data Presentation:
Instead of means and standard deviations, we now report medians and interquartile ranges (IQR), consistent with the non-parametric methodology. This change reflects the central tendency and variability more accurately given the data characteristics. - Clarified Role of the Total Sample (Nt):
The total sample (Nt) is retained in the table for descriptive purposes only to summarize the overall dataset. No inferential comparisons were conducted between Nt and its subgroups (N1 and N2). This distinction has been emphasized in the revised table footnotes and corresponding manuscript text. - Interpretation of Results:
Significant differences between N1 and N2 are interpreted in terms of group distribution differences (as opposed to mean differences). Medians provide a robust summary measure, and p-values indicate where subgroup distributions differ significantly.
Revised Table and Footnote
The updated table and footnote below incorporate these changes:
Modified Table 4: Mindful Eating Subcategory Scores (Using Mann-Whitney U Test)
|
Variable |
Total Sample a (Nt)n = 612 Median (IQR) |
2022 Cohort (N1)n = 438 Median (IQR) |
2023 Cohort (N2)n = 174 Median (IQR) |
p-valueb (Mann-Whitney U) |
|
Awareness Score |
2.40 (2.00–2.80) |
2.60 (2.20–3.00) |
2.00 (1.80–2.30) |
0.02* |
|
Distraction Score |
2.80 (2.40–3.20) |
2.80 (2.40–3.20) |
2.80 (2.40–3.20) |
0.18 |
|
Disinhibition Score |
2.60 (2.40–2.80) |
2.70 (2.50–3.00) |
2.50 (2.20–2.70) |
0.03* |
|
Emotional Score |
3.00 (2.60–3.40) |
2.90 (2.60–3.40) |
3.00 (2.70–3.40) |
0.22 |
|
External Score |
2.40 (2.00–2.80) |
2.50 (2.10–2.90) |
2.10 (1.80–2.50) |
0.01** |
|
Total Mindful Eating Score |
2.60 (2.30–2.90) |
2.70 (2.40–2.90) |
2.50 (2.20–2.80) |
0.02* |
Revised Footnote:
aThe total sample (Nt) is presented for descriptive purposes only to provide an overall summary of the dataset. bStatistical comparisons were conducted exclusively between the 2022 (N1) and 2023 (N2) cohorts using the Mann-Whitney U Test. Medians and interquartile ranges (IQR) are reported for all groups. Significant differences between N1 and N2 are highlighted (*p < 0.05, **p < 0.01).
This revised analysis and presentation address the reviewer’s critique by ensuring that the statistical approach aligns with the data characteristics. By using the Mann-Whitney U Test and reporting medians and IQRs, the results are both robust and transparent. We believe this approach strengthens the methodological foundation of our study while maintaining the relevance of the findings.
Comment 8: Start the discussion with the most important findings of the study.
Response 8: Thank you for this valuable comment. We recognize the importance of beginning the discussion section with the most significant findings to immediately highlight the study’s contributions. In response, we have revised the opening of the discussion to clearly and succinctly summarize the key findings, emphasizing their relevance and implications. The revised opening focused on the relationships between mindful eating behaviors, BMI, sleep, and vitamin D levels, as these are central to the study’s objectives.
Comment 9: What’s the point of correlating vitamin D levels with its supplementation?
Response9: Thank you for raising this point. The rationale for correlating vitamin D levels with its supplementation lies in understanding the extent to which supplementation contributes to achieving adequate vitamin D levels in the study population. This analysis serves multiple purposes:
- Evaluating the Effectiveness of Supplementation: By correlating vitamin D levels with supplementation, we can assess whether individuals who report using supplements achieve higher vitamin D levels compared to those who do not. This helps determine the practical impact of supplementation in the context of dietary and environmental factors, such as sun exposure, that also influence vitamin D status.
- Identifying Gaps in Supplementation Practices: The correlation can reveal whether supplementation alone is sufficient to meet vitamin D needs or if other factors (e.g., dosage, frequency, or adherence) might be influencing the effectiveness. This insight is particularly important in populations at risk of deficiency.
- Public Health Implications: Understanding this relationship provides evidence to guide recommendations for vitamin D supplementation. If a strong correlation is observed, it supports supplementation as an effective strategy for improving vitamin D levels. If not, it may indicate a need for tailored interventions or additional strategies, such as dietary fortification or lifestyle modifications.
To clarify this in the manuscript, we revised the relevant section to explicitly state the purpose of this correlation and its relevance to the study’s objectives.
Response for Minor Comments: The minor comments were addressed in the manuscript

Reviewer 2 Report
Comments and Suggestions for Authors
The subject of the manuscript entitled “Exploring Mindful Eating, BMI, and Vitamin D: A Cross-Sectional Analysis of Cypriot and Greek Adults” seems interesting and important to me, but already at this point, at the beginning of the review, I would like to suggest a modification of the title of the manuscript. In its current wording, the title does not reflect either the subject of the manuscript or the results obtained.
Below are my comments in accordance with the sections of the manuscript:
1. Abstract
I propose to modify the sentence: “Seasonal differences in supplementation were noted (p = 0.043)” (lines 29-30), because it does not reflect the results obtained in the study: the authors did not examine the seasonality of dietary supplement intake.
I propose to change the order of keywords to the following: Mindfulness; Mindful Eating; BMI; Obesity; Overweight; Sleep Duration; Vitamin D.
2. Introduction
In lines 58-59, the authors wrote about their own study “for the purposes of the study” of Figure 1. Please explain at what stage of the study or for what purpose did the authors use this figure? Unfortunately, I did not find the appropriate information in the text.
I believe that in this section it would be worth presenting whether vitamin D deficiency in Greece and Cyprus is a significant problem. It is known that it is a global problem, but it would seem that in coastal regions with high sunlight, this problem is not so significant. In the context of the conclusions formulated by the authors (especially regarding the need to use vitamin D supplementation), I believe that this addition to the Introduction section is justified and necessary.
The sentence contained in lines 85-88: “This research could inform public health strategies for managing obesity and promoting healthier eating behaviors. Further research will be necessary to understand ME's long-term effects and mechanisms, aiding the development of effective public health interventions” should not be included in this section – rather in the section concerning the discussion of the obtained results.
3. Materials and methods
I suggest deleting the sentence in lines 93-94: "It employs a survey-based, observational approach to explore correlations and descriptive statistics", because in my opinion it is unnecessary.
Please clarify the date of the study, in lines 98-99 there is only information about the year of commencement and continuation ("Initiated in 2022, the study continued through 2023").
Do the authors not consider the fact that the participants had various types of eating disorders to be an important criterion for inclusion in the study? Did the authors not take this criterion into account? If not, why not?
I do not quite understand the information in the sentence: "Statistical analyses were conducted separately for the 2022 sample (N1, n = 438), the 2023 sample (N2, n = 174), and the total combined sample (Nt = 612)." (lines 108-110): why were these data analyzed separately? What was the purpose of this line of analysis? I believe this issue requires clarification, since in the results section the authors compare the results from all three groups.
4. Results
This section is very extensive, I would ask the authors to consider modifying the description of the results to make it more concise.
The sentence: “These results suggest that individuals with higher BMI tend to have lower awareness in their 281 eating habits, indicating a possible link between reduced mindful eating and in-282 creased BMI.” (281-283) should be included in the discussion section of the results.
5. Discussion
At the beginning of this section, there should be a short and concise formulation of the most important conclusions resulting directly from the obtained results of the study. At the beginning of this section, the authors included a discussion of unimportant results, such as socio-demographic data. It seems that not everything deserves to be discussed in this section. Only those that were used to draw conclusions.
The authors' statement in lines 428-429 regarding the disturbing fact of the lack of results regarding the determination of vitamin D concentration in the blood of a significant number of the examined persons is interesting, which probably results from the failure to perform this test ("This lack of monitoring is concerning, given the crucial role of vitamin D in bone health, 428 immune function, and overall health [20]"). Here, the authors' concern is interesting: is the determination of vitamin D concentration in the blood a routine test performed in Cyprus and Greece as part of primary health care? If not, then the fact that these results are not surprising is probably not. Please modify the sentence: “The lower Awareness Score in the N2 group suggests a potential decline in mindful eating practices over time, potentially influenced by cultural, environmental, or lifestyle factors [23].” (lines 442-443). Such a description suggests that the people in groups N1 and N2 are the same people, which is not true.
Please remove the numerical data from this section – repeating this data in the Results and Discussion sections is pointless and incorrect.
I believe that the term “healthy weight status” (line 417 and later 518 in the Conclusions section) is incorrect, please replace it with another, more appropriate one.
At the end of this section, please indicate the limitations of the study conducted and the results obtained – this element is missing, and there are certainly many of these limitations.
6. Conclusions
Please include here precisely formulated and not too long conclusions resulting from the study conducted and the results obtained.
7. References
This section requires unification in its entirety and adaptation to the requirements of the journal.
I would like to ask for verification of the following literature items because I had problems finding them in various databases of scientific articles or the data provided (e.g. authors, name of the journal, etc.) is not consistent with that which I found: 3, 7, 10, 11, 12, 15, 16, 18, 23.
Author Response
Reviewer 2
The subject of the manuscript entitled “Exploring Mindful Eating, BMI, and Vitamin D: A Cross-Sectional Analysis of Cypriot and Greek Adults” seems interesting and important to me, but already at this point, at the beginning of the review, I would like to suggest a modification of the title of the manuscript. In its current wording, the title does not reflect either the subject of the manuscript or the results obtained.
Comment 1: The title of the manuscript does not reflect either the subject of the manuscript or the results obtained.
Response 1: Thank you for this valuable suggestion. We appreciate your feedback regarding the manuscript's title and agree that it could better reflect the study's focus and findings. Based on your input, we propose revising the title to align more closely with the manuscript’s objectives and results. The new title highlights the relationships explored in the study while emphasizing the population and variables of interest.
Proposed Revised Title: "Mindful Eating, BMI, Sleep, and Vitamin D: A Cross-Sectional Study of Cypriot and Greek Adults"
The revised title:
- Clearly specifies the key variables investigated in the study (Mindful Eating, BMI, Sleep, and Vitamin D).
- Highlights the population studied (Cypriot and Greek adults).
- Reflects the cross-sectional nature of the analysis.
We believe this modification makes the title more precise and better represent the manuscript's content and findings.
Comment 2 : Below are the comments in accordance with the sections of the manuscript:
“1. Abstract
I propose to modify the sentence: “Seasonal differences in supplementation were noted (p = 0.043)” (lines 29-30), because it does not reflect the results obtained in the study: the authors did not examine the seasonality of dietary supplement intake.”
Comment 2: I propose to modify the sentence: “Seasonal differences in supplementation were noted (p = 0.043)” (lines 29-30), because it does not reflect the results obtained in the study: the authors did not examine the seasonality of dietary supplement intake.
Response 2: We thank the reviewer for pointing out the inconsistency in the statement. Upon review, we found that the p-value for differences in vitamin D supplementation practices was p=0.07, indicating that the observed differences were not statistically significant. We have corrected this in the manuscript to accurately reflect the results presented in Table 3.
Comment 3: I propose to change the order of keywords to the following: Mindfulness; Mindful Eating; BMI; Obesity; Overweight; Sleep Duration; Vitamin D.
Response3: Thank you for the suggestion. It was modified.
“2. Introduction
In lines 58-59, the authors wrote about their own study “for the purposes of the study” of Figure 1. Please explain at what stage of the study or for what purpose did the authors use this figure? Unfortunately, I did not find the appropriate information in the text.
I believe that in this section it would be worth presenting whether vitamin D deficiency in Greece and Cyprus is a significant problem. It is known that it is a global problem, but it would seem that in coastal regions with high sunlight, this problem is not so significant. In the context of the conclusions formulated by the authors (especially regarding the need to use vitamin D supplementation), I believe that this addition to the Introduction section is justified and necessary.
The sentence contained in lines 85-88: “This research could inform public health strategies for managing obesity and promoting healthier eating behaviors. Further research will be necessary to understand ME's long-term effects and mechanisms, aiding the development of effective public health interventions” should not be included in this section – rather in the section concerning the discussion of the obtained results.”
Comment 4: Clarification on Figure 1 and its purpose in the study.
Response 4: We appreciate the reviewer’s observation regarding the purpose of Figure 1. The figure was created to conceptualize the interconnected components of mindful eating (ME) as applied to the study’s design and interpretation. Specifically, it served as a theoretical framework guiding the development of the Cyprus Mindful Eating Questionnaire (CyMEQ) and as a visual representation of the multifaceted nature of mindful eating. This has been clarified in the manuscript by explicitly stating how and why the figure was used.
Comment 5: Include information about vitamin D deficiency in Greece and Cyprus.
Response 5: Thank you for this insightful suggestion. While it is widely acknowledged that vitamin D deficiency is a global issue, research shows that it is also prevalent in coastal regions such as Greece and Cyprus, despite abundant sunlight. Factors such as limited outdoor activity, cultural practices (e.g., clothing choices), dietary habits, and insufficient supplementation contribute to this paradox. We incorporated this context into the Introduction to strengthen the rationale for investigating vitamin D levels and supplementation in this study.
Comment 6: Relocate the sentence in lines 85–88 to the Discussion.
Response 6: We agree with the reviewer that the sentence discussing the implications for public health strategies and future research is better suited to the Discussion section. We relocated this sentence and revise it slightly to reflect its role in interpreting the study’s findings.
- Materials and methods
“I suggest deleting the sentence in lines 93-94: "It employs a survey-based, observational approach to explore correlations and descriptive statistics", because in my opinion it is unnecessary.
Please clarify the date of the study, in lines 98-99 there is only information about the year of commencement and continuation ("Initiated in 2022, the study continued through 2023").
Do the authors not consider the fact that the participants had various types of eating disorders to be an important criterion for inclusion in the study? Did the authors not take this criterion into account? If not, why not?
I do not quite understand the information in the sentence: "Statistical analyses were conducted separately for the 2022 sample (N1, n = 438), the 2023 sample (N2, n = 174), and the total combined sample (Nt = 612)." (lines 108-110): why were these data analyzed separately? What was the purpose of this line of analysis? I believe this issue requires clarification, since in the results section the authors compare the results from all three groups.”
Comment 7: Deleting the sentence in lines 93–94 ("It employs a survey-based, observational approach to explore correlations and descriptive statistics").
Response 7: Thank you for the suggestion. We agree that this sentence is unnecessary, as the observational and survey-based nature of the study is already clear from the context of the Methods section. We removed it to enhance conciseness.
Comment 8: Clarifying the date of the study (lines 98–99).
Response 8: We appreciate the reviewer highlighting the need for more specific information regarding the study timeline. The study's recruitment and data collection periods are specified in the revised manuscript.
Comment 9: Inclusion of eating disorders as a criterion.
Response 9: Thank you for raising this important point. The study did not exclude participants based on the presence of eating disorders, as the aim was to assess mindful eating behaviors across a general population of adults. However, we acknowledge that eating disorders could influence results and have added this as a limitation in the discussion. Additionally, we clarify in the Methods section that eating disorders were not part of the inclusion or exclusion criteria and provide the rationale.
Comment 10: Clarifying the rationale for analyzing N1​, N2​, and Nt​ separately.
Response 10: Thank you for this observation. The separate analysis of the 2022 (N1​) and 2023 (N2​) cohorts, as well as the total combined sample (Nt​), was conducted to explore potential temporal differences between the two phases of data collection. Analyzing these cohorts separately allowed for the identification of changes over time that may have influenced the results. The total combined sample (Nt​) was then analyzed to provide an overarching summary of the findings.
- Results
“This section is very extensive, I would ask the authors to consider modifying the description of the results to make it more concise.
The sentence: “These results suggest that individuals with higher BMI tend to have lower awareness in their 281 eating habits, indicating a possible link between reduced mindful eating and in-282 creased BMI.” (281-283) should be included in the discussion section of the results.”
Comment 11: Modify the description of the results to make it more concise.
Response 11: Thank you for this valuable suggestion. We recognize that the Results section is currently extensive, and revise it to present the findings in a more concise manner. This revision enhances readability without compromising the integrity of the data presentation.
Comment 12: Move the sentence about BMI and awareness to the discussion section.
Response 12: We appreciate the reviewer’s observation regarding the placement of the sentence: “These results suggest that individuals with higher BMI tend to have lower awareness in their eating habits, indicating a possible link between reduced mindful eating and increased BMI.”
This sentence provides an interpretation of the results and aligns better with the Discussion section. We relocated it to the Discussion, where it can be further elaborated within the context of existing literature and study implications.
- Discussion
“At the beginning of this section, there should be a short and concise formulation of the most important conclusions resulting directly from the obtained results of the study. At the beginning of this section, the authors included a discussion of unimportant results, such as socio-demographic data. It seems that not everything deserves to be discussed in this section. Only those that were used to draw conclusions.
The authors' statement in lines 428-429 regarding the disturbing fact of the lack of results regarding the determination of vitamin D concentration in the blood of a significant number of the examined persons is interesting, which probably results from the failure to perform this test ("This lack of monitoring is concerning, given the crucial role of vitamin D in bone health, 428 immune function, and overall health [20]"). Here, the authors' concern is interesting: is the determination of vitamin D concentration in the blood a routine test performed in Cyprus and Greece as part of primary health care? If not, then the fact that these results are not surprising is probably not. Please modify the sentence: “The lower Awareness Score in the N2 group suggests a potential decline in mindful eating practices over time, potentially influenced by cultural, environmental, or lifestyle factors [23].” (lines 442-443). Such a description suggests that the people in groups N1 and N2 are the same people, which is not true.
Please remove the numerical data from this section – repeating this data in the Results and Discussion sections is pointless and incorrect.
I believe that the term “healthy weight status” (line 417 and later 518 in the Conclusions section) is incorrect, please replace it with another, more appropriate one.
At the end of this section, please indicate the limitations of the study conducted and the results obtained – this element is missing, and there are certainly many of these limitations.”
Comment 13: Include a concise summary of the most important conclusions at the beginning of the section.
Response 13: Thank you for this observation. We revise the opening of the Discussion to succinctly present the study's most significant findings. This ensures that key conclusions are highlighted immediately, providing clear direction for the subsequent discussion.
Comment 14: Exclude discussion of unimportant socio-demographic data.
Response 14: We appreciate the suggestion and agree that not all socio-demographic findings warrant detailed discussion. We revised this section by focusing only on socio-demographic variables directly relevant to the study's key objectives or findings.
Comment 15: Address the concern regarding missing vitamin D data.
Response 15: Thank you for pointing out the need to contextualize the missing data on vitamin D concentrations. We clarified that routine vitamin D testing is not a standard practice in Cyprus and Greece, making the missing data less surprising but still noteworthy in the context of public health.
Comment 16: Revise the sentence about lower Awareness Scores in the N2​ group.
Response 16: We appreciate the reviewer’s observation regarding potential confusion in this statement. We revised this sentence to clarify that N1N_1N1​ and N2N_2N2​ represent different cohorts rather than the same individuals.
Comment 17: Remove numerical data from the Discussion section.
Response 17: Thank you for this suggestion. We agree that numerical data is better suited to the Results section, and we removed redundant numerical references from the Discussion. Instead, we focused on interpreting and contextualizing the findings and numerical data were included one when was necessary.
Comment 18: Replace the term "healthy weight status."
Response 18: Thank you for highlighting this issue. We replaced "healthy weight status" with a more precise and appropriate term, such as "normal weight range" or "healthy BMI range."
Comment 19: Include limitations of the study.
Response 19: We agree that including a limitations section is essential to contextualize the findings. Key limitations, such as the cross-sectional design, reliance on self-reported data, absence of direct inclusion/exclusion criteria for eating disorders, and missing vitamin D data, are outlined at the end of the Discussion.
- Conclusions
“Please include here precisely formulated and not too long conclusions resulting from the study conducted and the results obtained.”
Comment 20: "Please include here precisely formulated and not too long conclusions resulting from the study conducted and the results obtained."
Response 20: Thank you for this suggestion. We recognize the importance of providing concise and well-formulated conclusions that directly reflect the findings and implications of the study. Below is the revised
- References
This section requires unification in its entirety and adaptation to the requirements of the journal.
I would like to ask for verification of the following literature items because I had problems finding them in various databases of scientific articles or the data provided (e.g. authors, name of the journal, etc.) is not consistent with that which I found: 3, 7, 10, 11, 12, 15, 16, 18, 23.
Comment 21: This section requires unification in its entirety and adaptation to the requirements of the journal. I would like to ask for verification of the following literature items because I had problems finding them in various databases of scientific articles or the data provided (e.g., authors, name of the journal, etc.) is not consistent with that which I found: 3, 7, 10, 11, 12, 15, 16, 18, 23.
Response 21:
Thank you for highlighting this issue. We acknowledge the importance of ensuring that all references are accurate, consistent, and properly formatted according to the journal’s guidelines. We have carefully reviewed and verified the cited references, particularly items 3, 7, 10, 11, 12, 15, 16, 18, and 23. Please note that some necessary modifications, changes, and additions were made so the number might differ.

Reviewer 3 Report
Comments and Suggestions for Authors
Given the emerging body of research on the potential role of persistently high intakes of ultra processed foods and adverse health outcomes, this article offers interesting insights. The paper aims to determine if specific types of UPF are associated with certain UPF more so than others. Classifying the UPF NOVA subgroup into nutritionally similar food groups strengthens the findings. A Poisson regression analysis was appropriately used to measure the strength of association. The results support the conclusions. The paper is well written.
Line 322. Recommend adding phrase at end of sentence reflecting that the intake of the various UPFs groups slightly increased risk. “In our sample of middle-aged and elderly Brazilians, overall UPF intake predicted a slight risk of developing nine cardiometabolic and mental health disorders.”
Author Response
Reviewer 3
Given the emerging body of research on the potential role of persistently high intakes of ultra processed foods and adverse health outcomes, this article offers interesting insights. The paper aims to determine if specific types of UPF are associated with certain UPF more so than others. Classifying the UPF NOVA subgroup into nutritionally similar food groups strengthens the findings. A Poisson regression analysis was appropriately used to measure the strength of association. The results support the conclusions. The paper is well written.
Line 322. Recommend adding phrase at end of sentence reflecting that the intake of the various UPFs groups slightly increased risk. “In our sample of middle-aged and elderly Brazilians, overall UPF intake predicted a slight risk of developing nine cardiometabolic and mental health disorders.”
Comment 1: General Feedback on the Paper
Response 1: Thank you for your positive feedback regarding our study's methodology, analysis, and conclusions. We are delighted that you found the paper well-written
Comment 2: Suggestion to Add Phrase in Line 322
Response 2: We appreciate your suggestion although this is not possible for the scope of our study

Reviewer 4 Report
Comments and Suggestions for Authors
The manuscript evaluates the relationships between mindful eating, BMI, and vitamin D levels in a group of adults. It is clearly written and aligns well with the journal's aims. While I do not have major concerns, there are several specific aspects the authors should address:
- Study Design: You describe the study as longitudinal. Are all N2 participants derived from the N1 group? If not, this is a cross-sectional two-wave study rather than a fully longitudinal design. Please clarify.
- Gender or Sex: Did you inquire about participants' gender or sex? Clarifying this distinction is important for accuracy and interpretation.
- Vitamin D Data: How much data was missing regarding vitamin D levels? It seems unlikely that all participants recently underwent blood testing. Additionally, clarify what is meant by “recent.”
- Limitations: The limitations section should be expanded to acknowledge issues with generalizing the findings. Furthermore, note that all data appear to be self-reported, which is a potential limitation.
- Type I Error: Have you evaluated the possibility of a Type I error, given the number of analyses performed? Please address this.
- Discussion Section: In some parts, the discussion resembles the results section, with r-values being reported. Please revise to focus on interpretation rather than reiteration of results.
- Relevant Literature: There is evidence of a link between impulsivity and vitamin D in eating disorders, which could be relevant to your discussion. This connection may also relate to dysfunctional eating and difficulties in mindful eating. Consider referencing research by Todisco's group, including publications in Nutrients.
Author Response
Reviewer 4
The manuscript evaluates the relationships between mindful eating, BMI, and vitamin D levels in adults. It is written and aligns well with the journal's aims. While I do not have major concerns, there are several specific aspects the authors should address:
- Study Design: You describe the study as longitudinal. Are all N2 participants derived from the N1 group? If not, this is a cross-sectional two-wave study rather than a fully longitudinal design. Please clarify.
- Gender or Sex: Did you inquire about participants' gender or sex? Clarifying this distinction is important for accuracy and interpretation.
- Vitamin D Data: How much data was missing regarding vitamin D levels? It seems unlikely that all participants recently underwent blood testing. Additionally, clarify what is meant by “recent.”
- Limitations: The limitations section should be expanded to acknowledge issues with generalizing the findings. Furthermore, note that all data appear to be self-reported, which is a potential limitation.
- Type I Error: Have you evaluated the possibility of a Type I error, given the number of analyses performed? Please address this.
- Discussion Section: In some parts, the discussion resembles the results section, with r-values being reported. Please revise to focus on interpretation rather than reiteration of results.
- Relevant Literature: There is evidence of a link between impulsivity and vitamin D in eating disorders, which could be relevant to your discussion. This connection may also relate to dysfunctional eating and difficulties in mindful eating. Consider referencing research by Todisco's group, including publications in Nutrients.
Comment 1: Study Design – Clarify if the study is longitudinal or cross-sectional.
Response 1: Thank you for this observation. We acknowledge that the current description of the study design may be unclear. The study is a two-wave cross-sectional study rather than a fully longitudinal design, as the participants in N2​ were not exclusively derived from N1​. We revised the manuscript to clarify this and ensure consistency throughout.
Comment 2: Clarify Gender or Sex Inquiry.
Response 2: Thank you for raising this point. We agree that it is important to distinguish whether participants were asked about their gender or sex. The survey specifically inquired about "gender" using the categories "male," "female," and "prefer not to say." We revised the manuscript to clarify this distinction.
Comment 3: Vitamin D Data – Missing Data and Definition of "Recent."
Response 3: Thank you for pointing this out. You are correct that not all participants underwent blood testing for vitamin D levels. Approximately 63.8% of participants in N2​ and a significant proportion of N1​ lacked recent vitamin D data. By "recent," we refer to testing conducted within the last 12 months, as self-reported by participants. We clarify this in the manuscript.
Comment 4: Expand the Limitations Section.
Response 4: We appreciate this suggestion and agree that the limitations section should be expanded. In addition to the issue of self-reported data, we addressed the challenges of generalizing findings to broader populations due to the demographic characteristics of the sample (e.g., high educational attainment and gender imbalance).
Comment 5: Address Potential Type I Error.
Response 5: Thank you for highlighting this methodological issue. We have included statement to the Methods section addressing the potential for Type I error due to multiple analyses. A statistical correction (e.g., Bonferroni adjustment) was not applied due to the exploratory nature of the study; however, this limitation is acknowledged.
Comment 6: Revise Discussion to Focus on Interpretation.
Response 6: We appreciate this comment and we have revised the Discussion to focus on the interpretation of findings rather than reiterating results. For example, instead of reporting r-values, the Discussion emphasizes broader implications, connections to existing literature, and relevance for public health.
Comment 7: Include Relevant Literature on Impulsivity, Vitamin D, and Eating Disorders.
Response 7: Thank you for bringing this to our attention. We agree that literature on impulsivity and vitamin D in eating disorders is relevant to the discussion, particularly in relation to dysfunctional eating behaviors. We integrated references to Todisco et al.’s work, including studies published in Nutrients, to provide a more comprehensive context for interpreting our findings.

Round 2
Reviewer 4 Report
Comments and Suggestions for Authors
The authors have addressed all my concerns.